# Deep Dive into the Long Haul: Analysis of Symptom Clusters and Risk Factors for Post-Acute Sequelae of COVID-19 to Inform Clinical Care

**DOI:** 10.3390/ijerph192416841

**Published:** 2022-12-15

**Authors:** Nicole H. Goldhaber, Jordan N. Kohn, William Scott Ogan, Amy Sitapati, Christopher A. Longhurst, Angela Wang, Susan Lee, Suzi Hong, Lucy E. Horton

**Affiliations:** 1Department of Surgery, School of Medicine, University of California San Diego, La Jolla, CA 92093, USA; 2Division of Biomedical Informatics, Department of Medicine, School of Medicine, University of California San Diego, La Jolla, CA 92093, USA; 3Department of Psychiatry, School of Medicine, University of California San Diego, La Jolla, CA 92093, USA; 4Department of Pediatrics, School of Medicine, University of California San Diego, La Jolla, CA 92093, USA; 5Division of Pulmonology, Department of Medicine, School of Medicine, University of California San Diego, La Jolla, CA 92093, USA; 6Division of Rheumatology, Allergy, and Immunology, Department of Medicine, School of Medicine, University of California San Diego, La Jolla, CA 92093, USA; 7Herbert Wertheim School of Public Health and Human Longevity Science, University of California San Diego, La Jolla, CA 92093, USA; 8Division of Infectious Diseases and Global Public Health, Department of Medicine, School of Medicine, University of California San Diego, La Jolla, CA 92093, USA

**Keywords:** Long COVID, post-acute sequelae, symptom cluster, neurocognitive

## Abstract

Long COVID is a chronic condition characterized by symptoms such as fatigue, dyspnea, and cognitive impairment that persist or relapse months after an acute infection with the SARS-CoV-2 virus. Many distinct symptoms have been attributed to Long COVID; however, little is known about the potential clustering of these symptoms and risk factors that may predispose patients to certain clusters. In this study, an electronic survey was sent to patients in the UC San Diego Health (UCSDH) system who tested positive for COVID-19, querying if patients were experiencing symptoms consistent with Long COVID. Based on survey results, along with patient demographics reported in the electronic health record (EHR), linear and logistic regression models were used to examine putative risk factors, and exploratory factor analysis was performed to determine symptom clusters. Among 999 survey respondents, increased odds of Long COVID (n = 421; 42%) and greater Long COVID symptom burden were associated with female sex (*OR* = 1.73, 99% CI: 1.16–2.58; β = 0.48, 0.22–0.75), COVID-19 hospitalization (*OR* = 4.51, 2.50–8.43; β = 0.48, 0.17–0.78), and poorer pre-COVID self-rated health (*OR* = 0.75, 0.57–0.97; β = −0.19, −0.32–−0.07). Over one-fifth of Long COVID patients screened positive for depression and/or anxiety, the latter of which was associated with younger age (*OR* = 0.96, 0.94–0.99). Factor analysis of 16 self-reported symptoms suggested five symptom clusters—gastrointestinal (GI), musculoskeletal (MSK), neurocognitive (NC), airway (AW), and cardiopulmonary (CP), with older age (β = 0.21, 0.11–0.30) and mixed race (β = 0.27, 0.04–0.51) being associated with greater MSK symptom burden. Greater NC symptom burden was associated with increased odds of depression (*OR* = 5.86, 2.71–13.8) and anxiety (*OR* = 2.83, 1.36–6.14). These results can inform clinicians in identifying patients at increased risk for Long COVID-related medical issues, particularly neurocognitive symptoms and symptom clusters, as well as informing health systems to manage operational expectations on a population-health level.

## 1. Introduction

COVID-19 was initially described as an acute respiratory illness, but now it is understood that the severe acute respiratory syndrome coronavirus 2 (SARS-CoV-2) virus can cause a broad spectrum of clinical illness across multiple organ systems. The disease affects patients following the initial phase of infection and has, therefore, been termed “post-COVID syndrome”, “post-acute COVID-19 sequelae”, “Post-Acute Sequelae of SARS-CoV-2 infections (PASC)”, and perhaps most commonly, “Long COVID”. Symptoms of Long COVID may be new onset, following initial apparent recovery from an acute SARS-CoV-2 infection episode, or may persist from the time of the initial illness [1,2,3,4]. Symptoms may also fluctuate or relapse over time. Out of more than 200 unique symptoms reported in confirmed or suspected cases, the most common include fatigue, shortness of breath, chest tightness, and cough, which may negatively impact daily functioning through both decreased quality of life and ability to work [3,5,6,7,8,9]. 

Long COVID appears to have a neurocognitive component [10,11,12,13], in addition to the wide range of physical symptoms, with emerging studies reporting an elevated incidence of mood disorders such as depression (>15%) and anxiety (>20%) [14,15], which increase in prevalence over time following the original infection [10]. According to the World Health Organization (WHO), in 2019 approximately one in eight people (12.5%) around the globe were living with a mental health disorder, most commonly depression or anxiety [16]. In 2020, with the onset of the COVID-19 pandemic, the prevalence of these conditions increased globally, with estimates as high as 28% and 26%, respectively, for depression and anxiety [17]. 

The true incidence of Long COVID remains unknown, with estimates ranging from 5% to over 50% of patients experiencing chronic sequelae after acute COVID-19 infections, depending on the populations investigated and/or the methods utilized to conduct these investigations [2,9,18]. Early studies have investigated the spectrum of Long COVID symptomatology primarily in patients who were hospitalized [7,19,20]; however, as diagnostic criteria for the illness continue to evolve, a full understanding of the clinical spectrum of Long COVID and underlying causes has yet to be achieved, particularly among the population that was not hospitalized. To our knowledge, few studies to date have sought to characterize and identify clusters of Long COVID symptoms, which perpetuates ambiguity in diagnosis and treatment plans, as well as in discerning population-level risk factors for developing Long COVID. Although a recent report of a large sample of patients seeking care in the United Kingdom describes certain demographic characteristics (i.e., older age, female sex, White race) and health conditions (i.e., pre-existing physical and mental disorders, obesity, and asthma) as key risk factors for Long COVID [18], symptom-specific associations with risk factors remain largely unknown.

In this study, we conducted a detailed analysis of population-based, self-reported survey data from hospitalized and non-hospitalized patients who had previously tested positive for COVID-19, which aimed to (i) identify sociodemographic and clinical factors that may be associated with increased risk for Long COVID and its correlates, (ii) assess the extent to which Long COVID symptom presentation has an underlying factor structure (i.e., symptom clusters), and (iii) determine whether risk for clinically meaningful anxiety and depression is associated with specific features of Long COVID symptomatology.

## 2. Materials and Methods

### 2.1. Study and Electronic Survey Design

An electronic survey was sent to patients aged 18 years or older in the University of California San Diego Health (UCSDH)’s electronic health record (EHR) who tested positive by nucleic acid amplification testing for COVID-19 between 1 March 2020, and 1 July 2021, and were still alive at the time of the initial survey outreach according to UCSDH records. In the survey, patients responded ‘No’, ‘Yes’, or ‘Maybe’ to whether they were “continuing to experience symptoms that [they] believe are caused by having COVID-19.” Patients who responded ‘Yes’ or ‘Maybe’ were categorized as having Long COVID in downstream analyses and administered symptom-specific survey items. 

Symptom-specific survey items asked patients “in the past 2 weeks, how often have you experienced the following symptom?” and were recorded on a five-point response scale (0 = never; 1 = rarely; 2 = sometimes; 3 = often; 4 = always), with higher scores indicating more frequent or severe symptomatology. The internal consistency of the sixteen-item symptom-specific component of the survey was high (McDonald’s Omega coefficient = 0.93). Patients were considered to have experienced a discrete symptom if reporting ‘sometimes’, ‘often’, or ‘always’ on the five-point scale. Patients were also asked to respond to two five-point items (1 = very poor; 2 = poor; 3 = fair; 4 = good; 5 = excellent) self-rating their health both before and after having had COVID-19. 

Sociodemographic information, including age, sex, race, and Healthy Places Index (HPI), were derived from EHR. The HPI is a scoring system developed by the Public Health Alliance of Southern California that incorporates 25 community characteristics known to drive health and life expectancy spanning the eight following weighted domains: economic (0.32), education (0.19), healthcare access (0.05), housing (0.05), neighborhood (0.08), clean environment (0.05), social (0.10), and transportation (0.16). Each census tract has a unique score that is adjusted to a percentile between 0 and 100, with a higher percentile indicating healthier community conditions. The population data for each census tract are taken from the Decennial census of 2010 [21]. The institutional EHR was also utilized to confirm whether patients had been admitted to the hospital or treated with monoclonal antibodies during their acute COVID-19 infection.

### 2.2. Depression and Anxiety Screening Questionnaires

Included in the electronic survey were the Generalized Anxiety Disorder 2-item (GAD-2) and the Patient Health Questionnaire 2-item (PHQ-2) to screen for the presence of clinically relevant anxiety and depressive symptomatology, respectively. Previously established threshold scores of ≥3 on either the GAD-2 or PHQ-2 were used to classify patients at putative high-risk for anxiety or depressive disorders.

### 2.3. Statistical Analysis

R 4.0.2 was utilized for all analyses. Sociodemographic and health-related characteristics were summarized using means and standard deviations to examine differences between patients reporting Long COVID symptoms and those without. Two-sample independent t-tests for continuous variables and Fisher’s exact or chi-squared tests for categorical variables were used to formally test the null hypotheses that no differences existed in sociodemographic and health-related variables between patients reporting Long COVID symptoms and those without. Multivariate logistic regression was used to test for differential covariate-adjusted odds of reporting Long COVID symptoms. Long COVID symptom burden was operationalized binarizing symptom frequency (1 = sometimes, often, always; 0 = never, rarely) and summing across 16 symptoms experienced within prior two weeks (possible range: 1–16). Ordinary least squares linear regression was used to test for associations between Long COVID symptom burden and sociodemographic and clinical variables. Effect sizes are reported where possible, with Cohen’s d ≈ 0.20 considered “small,” d ≈ 0.50 as “medium,” and d ≈ 0.80 as “large,” according to Cohen. Confidence intervals (CI) of 99% are reported for effect size estimates, test statistics, and estimated marginal means, unless otherwise indicated, to minimize Type I error due to multiple hypothesis testing. The sjPlot package was used to generate R2 and output tables [16]. Polychoric correlation coefficients were generated to visually assess correlations across COVID-19 related symptom frequencies (Appendix A). An exploratory factor analysis (EFA) of symptom frequency data among patients reporting Long COVID was conducted to determine whether the symptoms could be described by distinct clusters (i.e., factors). To determine the appropriate number of factors to extract, the polychoric correlation matrix for all 16 symptoms was subjected to a parallel analysis for FA using 1000 random draws, which indicated a five-factor solution best fit the data. EFA using five factors was performed based on polychoric correlation coefficients and maximum likelihood estimation. Oblimin rotation was performed to allow for correlations among factors, and items loading |>0.40| were retained in the final result. Individual factor scores were used as dependent variables in ordinal least-squares regression models to test whether sociodemographic and clinical features were associated with each symptom cluster, and logistic regression was used to test whether factor scores (i.e., cluster severity) were associated with differential odds of reporting elevated depression or anxiety symptoms.

## 3. Results

### 3.1. Participant Characteristics

The survey was sent to 9619 patients and achieved a 10.4% response rate. Preliminary results and analyses previously reported [22]. Across all respondents, an average of 331 days (range: 120–617; SD = 84) had elapsed between survey completion and the date of their SARS-CoV-2-positive test result.

Of the 999 respondents, 406 (41%) identified as male, 592 (59%) as female, and one (0.1%) as other. The average age of respondents was 51.5 years (range: 18–89 years), and 525 (53%) identified as White, 229 (23%) as Other Race or Mixed Race, 91 (9%) as Asian, 44 (4%) as Black or African American, five (0.5%) as American Indian or Alaska Native, 5 (0.5%) as Native Hawaiian or Other Pacific Islander, and 100 (10%) were unknown. Of the 962/999 with HPI data, the percentile average for respondents was 57.0 (range: 2–99), with 150 (16%) in the first quartile, 209 (22%) in the second quartile, 313 (33%) in the third quartile, and 290 (30%) in the fourth quartile, which was generally representative of San Diego County (mean = 54.7; first quartile: 18%; second quartile: 25%; third quartile: 31%; fourth quartile: 27% [23]).

### 3.2. Incidence and Sequelae of Long COVID Symptomatology

Almost half (421, 46.3%) of the respondents replied “yes” or “maybe” to currently having symptoms believed to be caused by having COVID-19 (i.e., Long COVID), and 409 provided responses to symptom-specific questions. Among those 409 individuals, weakness/tiredness (77.8%), sleep disturbances (67.2%), and difficulty thinking/concentrating (“brain fog”) (64.3%) were the most frequently reported of the 16 symptoms surveyed, followed by joint pain (60.4%), muscle aches (56.5%), headache (50.9%), reflux/frequent throat clearing (48%), difficulty breathing (35.9%), cough (34.5%), increased heart rate/palpitations (34.5%), loss of taste/smell (26.4%), diarrhea (23.7%), chest pain (23.5%), abdominal pain (22.5%), nausea/vomiting (15.2%) and fever (5.4%) (Figure 1). Of those experiencing Long COVID symptoms, 343 (83.9%) reported ≥3 symptoms, with a median of six symptoms (mean = 6.3; SD = 3.7). Over half of the patients reporting Long COVID symptoms (n = 216; 52.8%) reported some combination of absences from work/school (n = 75 of 216; 34.7%), disruption of daily activities (n = 143 of 216; 66.2%), or seeking medical care due to their symptoms (n = 123 of 216; 56.9%). Approximately one-quarter of patients experiencing symptoms of Long COVID reported being hospitalized due to COVID-19 (n = 97 of 421; 24.7%), and 45 (13.0%) reported having received monoclonal antibody treatment for SARS-CoV-2 infection. 

### 3.3. Sociodemographic and Clinical Predictors of Long COVID

Compared to patients reporting no chronic COVID-related symptoms, univariate analyses indicated that patients reporting Long COVID symptoms were 34% more likely to be women, approximately 4.5 times as likely to have been hospitalized for acute COVID-19 infection, and more than twice as likely to have been treated with intravenous monoclonal antibodies against SARS-CoV-2 (Table 1). In addition, patients experiencing Long COVID symptoms self-rated their overall health as poorer prior to their initial COVID-19 infection, and much poorer following infection relative to patients without Long COVID. An association was also found between monoclonal antibody use and poorer self-rated pre-COVID health (*X*^2^ = 17.1, df = 1, *p* < 0.001). Patients with Long COVID did not differ by age, race, or HPI compared to those without Long COVID (Table 1). Multivariate logistic regression analyses, adjusted for age, race, HPI, and monoclonal antibody administration, indicated that female sex (*OR* = 1.73, 99% CI: 1.16–2.58), acute COVID-19 hospitalization (*OR* = 4.51, 2.50–8.43), and poorer pre-COVID health (*OR* = 0.75, 0.57–0.97) were associated with increased risk of Long COVID. 

### 3.4. Sociodemographic and Clinical Correlates of Long COVID Burden of Disease 

Among patients reporting symptoms of Long COVID (n = 409), univariate analyses indicated that those with prior COVID-19 hospitalization reported a greater number of discrete symptoms (8.0, 99% CI: 7.0–8.9 symptoms) than those who were never hospitalized (6.0, 5.5–6.6 symptoms; t390 = 4.60, d = 0.54, 0.23–0.85). Multivariate analyses indicated that the number of discrete Long COVID symptoms reported (at a frequency of ‘sometimes’, ‘often’, or ‘always’) was positively associated with being female (βstd = 0.48, 99% CI: 0.22–0.75) and COVID-19-related hospitalization (βstd = 0.48, 0.17–0.78), and negatively associated with self-appraised health pre-COVID-19 infection (βstd = −0.19, −0.32–−0.07). Among patients with Long COVID symptoms, 21.7% and 22.6% of survey respondents scored ≥3 (i.e., positive) on the PHQ-2 and GAD-2 depression and anxiety screening questions, respectively, and previously hospitalized patients were also more likely to report clinically relevant depression (36.1% versus 17.6%; *OR* = 2.63, 99% CI: 1.34–5.16) and tended to have higher rates of anxiety (31.3% of hospitalized versus 19.7% non-hospitalized; OR = 1.85, 0.92–3.63). 

### 3.5. Long COVID Symptomatology Reflects Five Underlying Symptom Clusters

In order to explore whether distinct clusters of Long COVID symptomatology existed among those reporting symptoms and the extent to which symptom presentation patterns differed based on sociodemographic or clinical factors, including anxiety and depressive symptoms, EFA was performed. A five-factor solution accounted for 57% of the cumulative variance in symptomatology, with Factor 1 (gastrointestinal; GI cluster) predominantly loaded by nausea and vomiting, abdominal pain, diarrhea, headache, and fever (13.5% variance explained), Factor 2 (musculoskeletal; MSK cluster) loading by joint pain and muscle aches (12.1% variance explained), Factor 3 (neurocognitive; NC cluster) loading by brain fog, sleep disturbances, weakness, and tiredness (12.2%), Factor 4 (airway; AW cluster) loading by coughing and reflux (9.6%), and Factor 5 (cardiopulmonary; CP cluster) loading by chest pain, difficulty breathing, and heart palpitations (9.4%) (Figure 2). Only loadings >0.40 are shown and considered a significant load to a particular cluster. Of note, anosmia/ageusia had loadings of 0.07, 0.28, −0.09, −0.07, and −0.24, on the GI, neurocognitive, MSK, airway, and cardiopulmonary clusters, respectively. Linear regression analyses, adjusted for number of Long COVID symptoms, indicated that age was negatively associated with GI cluster scores (βstd = −0.09, 99% CI: −0.18–0.00) and positively associated with MSK cluster scores (βstd = 0.21, 99% CI: 0.11–0.30; Table 2). Patients of Other (or Mixed) race also had higher MSK cluster scores compared to White patients (βstd = 0.27, 99% CI: 0.04–0.51). Higher neurocognitive symptom scores tended to be associated with prior hospitalization for COVID-19 infection (βstd = 0.15, 99% CI: −0.02–0.33) and poorer self-rated health pre-COVID (βstd = −0.08, 99% CI: −0.18–0.01). Neither sex nor HPI were associated with symptom cluster scores.

### 3.6. Associations between Symptom Clusters, Anxiety, and Depression

Adjusted for age, sex, race, HPI, COVID-19 hospitalization, self-rated pre-COVID health, and number of symptoms, higher neurocognitive cluster scores were associated with greater odds of depressive symptoms (*OR* = 5.86, 99% CI = 2.71–13.78) and anxiety (*OR* = 2.83, 1.36–6.14; Table 3). Scores on other factors were not associated with differential odds of depressive symptoms or anxiety; however, older age was associated with lower odds of reporting anxiety (*OR* = 0.96, 0.96–0.99). Interestingly, women were nearly half as likely as men to report depressive symptoms (*OR* = 0.51, 0.21–1.19), though the association was not significant at *p* < 0.01.

## 4. Discussion

This study directly queried almost 10,000 patients who tested positive for COVID-19 infection at a UC San Diego Health (UCSDH) facility during the study period. Just under 1000 patients responded, with half (46.3%) reporting having one or more symptom consistent with Long COVID—aligning with the upper range of previous prevalence reports [14]. Weakness/tiredness, sleep disturbances, and difficulty thinking/concentrating (“brain fog”) were the most frequently reported symptoms, and most patients reported having an average of six symptoms out of the 16 provided (not including the additional free response “other” option). In this study, we performed deeper analyses to identify risk factors of increased odds of Long COVID and burden of individual symptoms as well as symptom clusters. 

Compared to patients who reported no Long COVID symptoms, we found that patients who reported experiencing Long COVID symptoms were more likely to be women, to have been hospitalized for acute COVID infection, and to have been treated with intravenous monoclonal antibodies. Patients who reported Long COVID symptoms additionally had poorer self-rated overall health (d = −0.29) prior to their initial COVID-19 infection, which is in line with a prior finding [18], and far poorer self-rated health following their initial COVID-19 infection (d = −1.21) relative to patients without Long COVID symptoms. Other studies have identified neurocognitive symptoms to be the most prevalent [4,24] and have identified women as being at greater risk of Long COVID than men [7,18,24]. The association with monoclonal antibody treatment is likely to reflect the fact that patients eligible for this treatment were at highest risk for severe COVID-19 infection and, therefore, likely have poor health and more comorbid conditions at baseline, which our study suggested. Our data on patient comorbidities were limited in this case, as a substantial portion of patients included do not receive their primary care at UCSDH but are in the system from obtaining ambulatory COVID testing at a UCSDH testing site, and full comorbidity data are not routinely collected during these interactions. 

Hospitalization due to acute SARS-CoV-2 infection correlated with a greater burden of chronic symptoms (n = 8 compared to 6 overall), indicating that severity of initial disease may correlate with burden of chronic Long COVID symptoms. This is also consistent with other recent studies [4,25], although earlier reports indicated incongruence between acute COVID symptom severity and development of Long COVID [8,26,27]. Perhaps earlier in the pandemic, patients who tested positive for COVID-19 progressed further into severe illness; however, severe disease was avoided as treatment and prevention approaches evolved (e.g., monoclonal antibodies, antiviral medications, vaccinations) and improved. It is important to note, however, that there is a latency difference between initial diagnosis and declaration of Long COVID symptoms in many of the relevant Long COVID studies conducted. This indicates the continued need for consistent and rigorous documentation and analyses of acute COVID symptoms and symptom severity to inform early detection and care of Long COVID. 

Multivariate analyses suggested that greater Long COVID symptom burden was associated with older age, female sex, as well as poorer self-appraised health prior to having COVID-19 and being hospitalized for COVID-19. Our findings are in general agreement with existing reports [18,28,29,30,31,32,33]. These results can inform clinicians to help identify patients at increased risk for Long COVID-related medical issues, as well as be used by health systems to manage operational expectations on a population-health level as there are implications for the application to clinical screening programs geared toward these specific patients [34]. Additionally, previously hospitalized patients were found to be almost twice as likely to report clinically relevant anxiety and depression, which also has operational implications for health systems and highlights the need to expand psychiatric services in collaboration with Long COVID centers to meet increased behavioral health needs. Accordingly, we found that >20% of patients who reported Long COVID symptoms scored positive on anxiety and depression screening questionnaires (GAD-2 and PHQ-2, respectively). 

This high burden of mood disorders is particularly challenging to address. Analysis of depression screening of the entire UC Health Primary Care patient population shows global worsening as a result of COVID-19 [23]. As has been discussed previously, it is difficult to distinguish the impact of psychological stress in response to COVID-19-related experiences from direct neuropathological effects of the SARS-CoV-2 virus in deciphering the etiology of elevated depressive mood [14]. In this study, we identify the sociodemographic, clinical, and symptom-associated factors that increase the risk for these mood disorders in a Long COVID patient cohort; however, we are unable to address this ambiguity. Other viewpoints suggest that there may be targetable biological mechanisms to help address these conditions such as neuroinflammation [35,36].

Factor analysis indicated five underlying symptom clusters of Long COVID, which we described as gastrointestinal (GI), neurocognitive (NC), musculoskeletal (MSK), airway (AW), and cardiopulmonary (CP)-related, respectively. The first cluster (GI), with the highest variance explained, contained primarily GI symptoms including nausea/vomiting, abdominal pain, and diarrhea with the highest levels in that group, which not unsurprisingly allocated together, as well as headache and fever at lower levels. The second cluster (MSK) included rheumatologic or musculoskeletal symptoms of joint pain and muscle aches. The third (NC) with primarily neurocognitive symptoms including brain fog, sleep disturbances, weakness/tiredness. The fourth (AW) with coughing and reflux, perhaps the most interesting combination. Additionally, the fifth (CP) with cardiopulmonary symptoms including chest pain, palpitations, and difficulty breathing. Among them, the two clusters that are first (GI and NC) represent the primary Long COVID symptom burden in the study population. Of note, the GI symptom cluster (which interestingly included non-GI symptoms of headache and fever) was more likely to be reported by younger responders, indicating that despite overall consensus in the current literature of older age being a risk factor for Long COVID symptoms, the need for more granular understanding of subpopulation-specific symptom clusters clearly exists. Older age and mixed race were, however, associated with increased MSK cluster presentation.

The discovery that symptom clusters identified in this analysis primarily correlate with physiologic systems suggests that Long COVID distributes itself on this basis, and that there are possibly distinct sub-types of Long COVID. Further research exploring the correlation of these symptom-based sub-types to specific symptoms during the initial acute infection might help identify underlying pathophysiology. In addition, it would be interesting to know if Long COVID sub-type correlates with patient comorbidities.

Reflux and coughing were found to align together in the same symptom cluster (AW), which is intriguing because in our experience treating patients in our UCSDH multidisciplinary Long COVID clinic, we have diagnosed laryngopharyngeal reflux in many patients presenting with cough. It has been hypothesized that reflux and coughing are correlative in other settings as well [37,38,39,40]. Factor 3, the cluster containing primarily neurocognitive symptoms (NC), was found to have the highest association with both positive anxiety and depression screens. Though causal inferences are not possible, this does suggest that neurocognitive symptoms, and the often-disabling impact of these symptoms, may contribute to poorer mental health.

Also interestingly, when risk factors were examined for the different symptom clusters, older adults had statistically significantly lower odds of reported anxiety. Older adults tend to have lower stress reactivity, and in general, better emotional regulation and well-being than younger adults [41,42,43]. School-based studies from around the globe report higher levels and incidence of anxiety in younger ages [44,45,46]. 

Some limitations for this study include its cross-sectional design. To that end, retrospective questions about pre-COVID-19 infection health status may be biased by current health status, particularly in Long COVID patients. Additionally, anxiety and depression screening questions are not surrogates for formal clinical diagnosis; although GAD-2 and PHQ-2 have good sensitivity and specificity for GAD and MDD, respectively. Psychiatric histories of these patients were not considered and were beyond the scope of this investigation. Further, it is also important to note that technology-based assessments can have inherent biases in terms of recruitment due to differences in literacy, linguistic inclusion, access to internet, and may under-represent individuals with high social vulnerability. Patient selection may also be biased as a quaternary academic medical center, which may result in recruitment of patients with higher underlying comorbidities that impact the analyses. However, it is important to note that the HPI quartile data in San Diego County was representative of our cohort data [47]. There are also limitations inherent to EHR data, as this can only be as accurate as what is entered into the system and by whom. While we wanted to include patient comorbidity and vaccination data, due to the nature of the survey and the population, appropriate data were not consistently available via EHR. Lastly, given the timing of this survey, the translational value of the findings may be limited, as infection cases from more recent variants, including Delta and Omicron, were not yet captured. Further investigation into inter- and intra-variant analysis is warranted.

## 5. Conclusions

The results of this study can inform clinicians in identifying patients at increased risk for Long COVID-related medical issues, particularly neurocognitive symptoms and symptom clusters. Further research about symptom clusters of Long COVID involving physiologic organ systems is needed and would enhance clinical understanding about different subtypes of Long COVID, thus better informing multidisciplinary care for this patient population. Additional studies focusing on risk factor identification, such as vaccination status and specific comorbidities, will also help improve screening for patients with Long COVID in the primary care setting. The relationship between symptom clusters, as well as SARS-CoV-2 variants and progression over time, would also be of interest in assessing the different manifestations of Long COVID and determining whether viral subtypes are associated with specific long-term sequelae. This study’s findings can guide future research and inform health systems in managing operational expectations on a population-health level to best care for patients experiencing Long COVID. 

## Figures and Tables

**Figure 1 ijerph-19-16841-f001:**
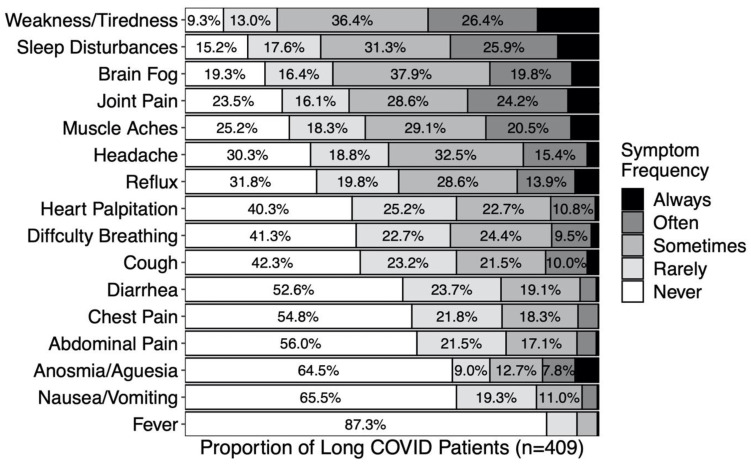
Proportion of patients reporting Long COVID symptoms at various frequencies over the preceding two weeks.

**Figure 2 ijerph-19-16841-f002:**
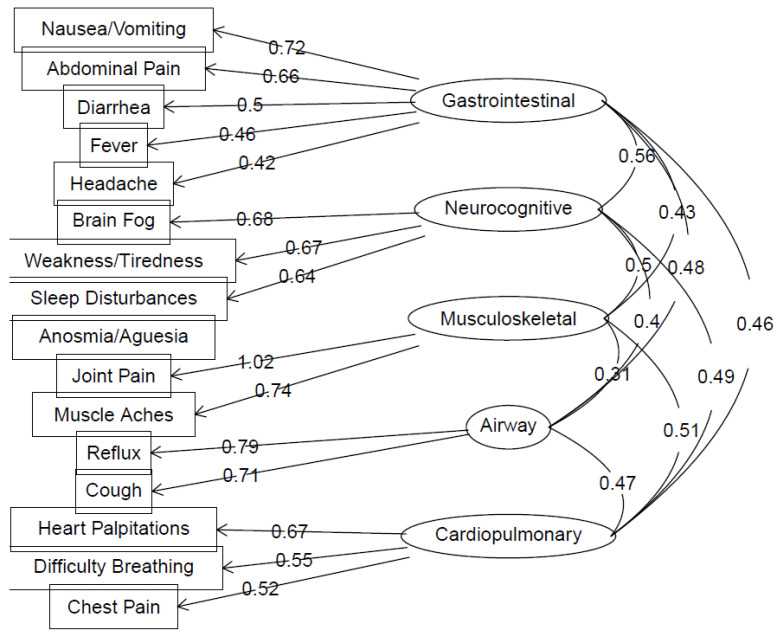
Factor loadings and intercorrelations obtained from exploratory factor analysis of Long COVID symptom frequency (16 items). Loadings are based on oblimin rotated solution of polychoric correlation matrix. Only loadings >0.40 are shown.

**Table 1 ijerph-19-16841-t001:** Participant sociodemographic and clinical characteristics.

	No Long COVID Symptoms Reported (n = 488)	Long COVID Symptoms Reported (n = 421)	Test Statistics (99% CI)
Age (years)	51.4 (16.3)	52.2 (15.0)	*d* = 0.05 (−0.12–0.22)
Race (%White/Black/AAPI/Other/NR)	56.8/4.7/9.2/19.7/9.6	50.8/4.0/10.7/24.7/9.7	*X*^2^_4_ = 4.80, *p* > 0.01
Sex (% Female)	56.4%	63.4%	***OR* = 1.34 (1.02–1.77)**
HPI (Percentile score)	58.0 (24.8)	56.6 (27.5)	*d* = −0.06 (−0.23–0.12)
Acute COVID-19 hospitalization	6.8%	24.7%	*OR* = 4.53 (2.59–8.22)
Pre-COVID self-rated health	4.27 (0.74)	4.04 (0.80)	***d* = −0.29 (−0.46–−0.12)**
Post-COVID self-rated health	4.24 (0.73)	3.25 (0.91)	***d* = −1.21 (−1.39–−1.02)**
Monoclonal antibody treatment	5.9%	11.5%	***OR* = 2.05 (1.06–4.07)**
Days since COVID+ result	315 (3.4)	348 (4.5)	***d* = 0.41 (0.23, 0.58)**

Means and standard deviation shown for continuous variables. Test statistics for continuous variables based on two-sample independent *t*-tests, converted to Cohen’s *d*, and for categorical variables based on chi-squared or Fisher’s exact tests, with 99% confidence intervals (CI) shown. Bold indicates that 99% CI does not cross 0 for continuous variables or 1 for categorical variables. Self-rated health based on five-point scale (1 = very poor, 2 = poor, 3 = fair, 4 = good, 5 = excellent). Higher Healthy Places Index (HPI) denotes higher resourced community. AAPI: Asian-American or Pacific Islander. NR: Not reported.

**Table 2 ijerph-19-16841-t002:** Sociodemographic and clinical predictors of symptom clusters.

	GI Cluster		MSK Cluster		NC Cluster		AW Cluster		CP Cluster	
*Predictors*	*Std. Beta*	*99% CI*	*Std. Beta*	*99% CI*	*Std. Beta*	*99% CI*	*Std. Beta*	*99% CI*	*Std. Beta*	*99% CI*
**Age**	−0.09 *	−0.18–0.00	0.21 ***	0.11–0.30	−0.03	−0.11–0.04	0.03	−0.06–0.12	−0.01	−0.08–0.07
**Sex [female]**	0.03	−0.16–0.21	−0.01	−0.21–0.19	0.11	−0.04–0.27	−0.09	−0.27–0.10	−0.02	−0.18–0.13
**Race [Black]**	0.06	−0.39–0.51	0.07	−0.42–0.57	−0.10	−0.48–0.29	0.04	−0.41–0.50	−0.14	−0.53–0.26
**Race [AAPI]**	−0.13	−0.41–0.15	0.20	−0.12–0.51	−0.12	−0.36–0.12	−0.1	−0.39–0.19	0.11	−0.14–0.36
**Race [Other]**	0.05	−0.17–0.26	0.27 ***	0.04–0.51	−0.01	−0.19–0.17	−0.15	−0.37–0.07	0.01	−0.18–0.19
**Race [NR]**	−0.02	−0.32–0.27	0.20	−0.12–0.53	0.04	−0.21–0.29	−0.26 *	−0.56–0.04	−0.01	−0.26–0.25
**HPI percentile**	−0.06	−0.14–0.03	−0.04	−0.14–0.05	0.00	−0.07–0.07	−0.01	−0.10–0.08	0.00	−0.08–0.07
**COVID−19 hospitalization**	0.04	−0.17–0.25	0.17	−0.07–0.40	0.15 *	−0.02–0.33	0.01	−0.20–0.23	0.11	−0.08–0.29
**Pre-COVID self-rated health**	−0.07	−0.18–0.03	−0.02	−0.14–0.10	−0.08 *	−0.18–0.01	0.01	−0.10–0.12	−0.01	−0.10–0.09
**No. of symptoms**	0.81 ***	0.72–0.90	0.69 ***	0.59–0.79	0.79 ***	0.71–0.87	0.72 ***	0.63–0.81	0.78 ***	0.70–0.86
**Observations**	377		377		377		377		377	
**R^2^/R^2^ adjusted**	0.638/0.629		0.569/0.559		0.701/0.694		0.562/0.551		0.682/0.674	

Standardized beta estimates with 99% confidence intervals (CI) shown for linear regression models. Self-rated health based on five-point scale (1 = very poor, 2 = poor, 3 = fair, 4 = good, 5 = excellent). Higher Healthy Places Index (HPI) denotes higher resourced community. AAPI: Asian-American or Pacific Islander. NR: Not reported. Number of symptoms based on binarization of symptom frequency (1 = sometimes, often, always; 0 = never, rarely) across 16 symptoms experienced within prior 2 weeks (range = 1–16). * *p* < 0.05; *** *p* < 0.005.

**Table 3 ijerph-19-16841-t003:** Sociodemographic, clinical, and symptom cluster predictors of depression and anxiety.

	Depression		Anxiety	
	*OR*	*99% CI*	*OR*	*99% CI*
** *Sociodemographics* **				
**Age**	0.98	0.95–1.01	0.96 **	0.94–0.99
**Sex [female]**	0.51 *	0.21–1.19	0.68	0.28–1.66
**HPI Percentile**	1.01	0.99–1.02	1.01	0.99–1.03
**Race [Black]**	1.76	0.24–9.95	0.77	0.05–5.26
**Race [AAPI]**	0.63	0.13–2.46	0.29 *	0.05–1.23
**Race [Other]**	1.37	0.52–3.61	1.09	0.42–2.76
**Race [NR]**	2.04	0.54–7.30	0.7	0.14–2.70
** *Clinical characteristics* **				
**COVID-19 hospitalization**	1.97 *	0.81–4.81	1.37	0.54–3.47
**Pre-COVID self-rated health**	0.76	0.47–1.22	0.69 *	0.42–1.11
**No. of Symptoms**	0.96	0.70–1.31	1.29 *	0.95–1.77
** *Symptom Clusters* **				
**GI**	0.93	0.50–1.72	0.62 *	0.33–1.15
**Neurocognitive**	5.86 ***	2.71–13.78	2.83 ***	1.36–6.14
**MSK**	0.99	0.55–1.77	0.77	0.41–1.40
**Airway**	0.91	0.46–1.78	0.83	0.42–1.58
**Cardiopulmonary**	0.89	0.44–1.80	1.19	0.60–2.40
**Observations**	377		375	
**R^2^ Tjur**	0.281		0.256	

Odds ratios (OR) with 99% confidence intervals (CI) shown for logistic regression models. Self-rated health based on five-point scale (1 = very poor, 2 = poor, 3 = fair, 4 = good, 5 = excellent). Higher Healthy Place Index (HPI) denotes higher resourced community. Symptom clusters based on factor analysis of frequency of 16 Long COVID symptoms. GI: gastrointestinal symptom cluster. MSK: musculoskeletal symptom cluster. AAPI: Asian-American or Pacific Islander. NR: Not reported. Number of symptoms based on binarization of symptom frequency (1 = sometimes, often, always; 0 = never, rarely) across 16 symptoms experienced within prior 2 weeks (range = 1–16). Tjur: Coefficient of determination, R^2^. ** p* < 0.05; ** *p* < 0.01; *** *p* < 0.001.

## Data Availability

The data presented in this study are available on request from the corresponding author.

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
