# Peer review of "Deep Dive into the Long Haul: Analysis of Symptom Clusters and Risk Factors for Post-Acute Sequelae of COVID-19 to Inform Clinical Care"

_ijerph, 2022, doi:10.3390/ijerph192416841_

Round 1
Reviewer 1 Report
General comments
The topic of the manuscript is highly relevant, as long COVID can cause different types and combinations of symptoms, and there is no definite test to diagnose these conditions. Consequently, it is essential to understand risk factors and assess symptom clusters for post-COVID syndrome. After a minor revision, this manuscript will provide an important contribution to this subject area.
The description of the problem context is comprehensive. The methodology is relevant, and the statistical techniques and their use are explained in exhaustive detail. However, some more information about the exploratory factor analysis would be beneficial, as noted in the specific comment below.
Specific comment
In section 3.2 and figure 2 (lines 237-262), a 5-factor solution is presented and explained. However, the symptom “anosmia/aguesia” is not included in any of the five factors. It would be beneficial to comment on this fact in the text.
Author Response
Dear IJERPH Editorial Office,
Thank you for the opportunity to revise our paper for IJERPH. We appreciate the time and efforts from the reviewers. We have addressed all of the comments offered by both reviewers in the revised manuscript and believe these changes have resulted in an improved manuscript. Below we have specifically addressed the comments (in italics) from both Reviewer 1 and 2, and attached is a revised manuscript with tracked changes as well as a clean copy of the updated version.
Reviewer 1: In section 3.2 and figure 2 (lines 237-262), a 5-factor solution is presented and explained. However, the symptom “anosmia/aguesia” is not included in any of the five factors. It would be beneficial to comment on this fact in the text.
Thank you for this comment. We have added a comment explaining why the symptom “anosmia/ageusia” is not included in the five factors (Lines 248-251). Only loadings >0.40 are considered a significant load to a particular cluster (and only these significant loading factors/clusters are shown in Figure 2). Please also note that there is a misspelling related to this change, and we have corrected “ageusia” to “ageusia” in the text and figure.
Reviewer 2: Although there could be more references in the background and discussion chapters, and I would suggest the conclusion chapter be supplemented with information regarding further research and why these are needed, I consider the article brings useful information to the researchers in the field.
Thank you for this insightful comment. As suggested, we have added 12 additional references to the background and discussion sections. Additionally, we have added several sentences to the conclusion section addressing the need for further research into specific aspects of Long COVID, including symptom clusters and sub-types of Long COVID (Lines 412-422).
Thank you again for your review of this work. We look forward to hearing your response.
Sincerely,
Lucy Horton, MD, MPH
Nicole Goldhaber, MD, MS
Reviewer 2 Report
This is an interesting article on the topic of symptoms of long-COVID, including neurocognitive ones. Long-COVID is still a debateble entity and I strongly believe that a cross-sectional, retrospective enquiry can produce bias. Results are clearly presented. Although there could be more references in the background and discussion chapters, and I would suggest the conclusion chapter be supplemented with information regarding further research and why these are needed, I consider the article brings useful information to the researchers in the field.
Author Response

(The authors gave the same response as above.)
